# Classification and Characterization of Tire-Road Wear Particles in Road Dust by Density

**DOI:** 10.3390/polym14051005

**Published:** 2022-03-02

**Authors:** Uiyeong Jung, Sung-Seen Choi

**Affiliations:** Department of Chemistry, Sejong University, 209 Neungdong-ro, Gwangjin-gu, Seoul 05006, Korea; dmldud3731@sju.ac.kr

**Keywords:** tire-road wear particle, road dust, density separation, characterization, asphalt pavement wear particle

## Abstract

Tire treads are abraded by friction with the road surface, producing tire tread wear particles (TWPs). TWPs combined with other particles on the road such as road wear particles (RWPs) and mineral particles (MPs), forming tire-road wear particles (TRWPs). Dust on an asphalt pavement road is composed of various components such as TRWPs, asphalt pavement wear particles (APWPs), MPs, plant-related particles (PRPs), and so on. TRWPs have been considered as one of major contaminants produced by driving and their properties are important for study on real abrasion behaviors of tire treads during driving as well as environmental contamination. Densities of the TRWPs are totally dependent on the amount of the other components deposited in the TWPs. In this study, a classification method of TRWPs in the road dust was developed using density separation and the classified TRWPs were characterized using image analysis and pyrolytic technique. Chloroform was used to remove APWPs from mixture of TRWPs and APWPs. TRWPs were found in the density range of 1.20–1.70 g/cm^3^. By decreasing the particle size of the road dust, the TRWP content in the road dust increased and its density slightly tended to increase. Aspect ratios of the TRWPs varied and there were many TRWPs with low aspect ratio below 2.0. The aspect ratio range was 1.2–5.2. Rubber compositions of the TRWPs were found to be mainly NR/SBR biblend or NR/BR/SBR triblend.

## 1. Introduction

Tire treads are abraded by friction between the road surface and the tire tread, resulting in the formation of tire-road wear particles (TRWPs) [1,2]. TRWPs are usually black particles with a stick or round shape, and, to some extent, they are encrusted by mineral and other tiny particles [3,4]. Since rubber has a slightly sticky surface [5], such particles can easily adhere to the rubber particle (tire tread wear particle, TWP) surface. When contaminants attach to the TWP surface, they can release to environment. Especially, the particles of small size float in the air to cause air pollution as fine dust or they enter the aquatic environment to cause water pollution [6]. Therefore, TRWPs have been considered a kind of microplastics and a pollutant [7,8].

In general, densities of the TWPs are about 1.2 g/cm^3^, while the TRWP densities should be higher than the TWPs due to the encrusted mineral particles [9,10]. Granite and marble are some of representative rock components and their densities are about 2.7 g/cm^3^ [11,12]. TRWPs with high density have many mineral particles or environmental contaminants attached to the surface. Since TRWPs have various densities, it is necessary to subdivide TRWPs by density and analyze their properties. Tire tread wear is also directly related to the lifespan of a tire, so careful and detailed researches on the abrasion behaviors are required. Real road dust is composed of various components including TRWPs, asphalt pavement wear particles (APWPs), mineral particles (MPs), glass particles (GPs), glass beads (GBs), plant-related particles (PRPs), road paint wear particles (RPWPs), plastic particles (PPs), and others [4]. Hence, TRWPs should be separated from the road dust at first to examine properties of TRWPs.

A density separation method has been employed for separation of TRWPs from the environmental samples, and it uses the density differences of the components. It was reported that a very high-density medium of sodium polytungstate solution (density: 2.2 g/cm^3^) was used for the density separation [1]. Using this method, the TRWPs were found in the floated fraction but lots of other particles also existed in the same fraction. Klockner et al. used an instrument consisting of a 50 mL volumetric flask equipped with PVC tube and sodium polytungstate solution of 1.9 g/cm^3^, and they could fractionate the sample 1 day after the solution treatment [2]. This method may be semi-automated using the instrument but it is somewhat complex and takes a long time. To effectively separate TRWPs from the environmental samples and determine their detailed densities, it is required to develop a new method. In this study, a separation method of TRWPs from the environmental sample of road dust was developed using salt solutions with different densities and chloroform. Chloroform was used to eliminate APWPs from the sample mixed with TRWPs. The TRWPs were classified by 0.10 g/cm^3^ density unit and their structural characteristics and rubber compositions were analyzed. We believe that this analytical method could be also applied to various environmental samples containing TRWPs such as sediments of river and sea.

## 2. Materials and Methods

### 2.1. Materials

Sodium bromide (purity: 99.0%) and sodium iodide (purity: 99.0%) were purchased from Ducksan Co. (Ansan, Korea). Chloroform (purity: 99.5%) was purchased from Samchun Chemical Co. (Seoul, Korea). Deionized water was used to prepare the salt solutions.

### 2.2. Matrix Solution Preparation for Density Separation

For density separation, salt solutions and chloroform were used. Sodium bromide and sodium iodide were used for the preparation of the salt solutions in deionized water. Density of the saturated sodium bromide solution was 1.52 g/cm^3^ at 22 °C. Sodium bromide solutions with densities of 1.20, 1.30, 1.40 and 1.50 g/cm^3^ were prepared. For the preparation of higher density solutions, sodium iodide was used. Density of the saturated solution of sodium iodide was 1.83 g/cm^3^ at 24 °C. Sodium iodide solutions with densities of 1.60 and 1.70 g/cm^3^ were prepared. Density of chloroform is 1.49 g/cm^3^.

### 2.3. Preparation of Road Dust Sample

Road dust was collected in January 2021 near a bus stop by sweeping with a broom. The road dust was separated by size (212–500, 106–212, and 63–106 μm) using a sieve shaker of Octagon 200 (Endecotts Co., London, UK). Morphology of the particles was analyzed using an image analyzer (EGVM 35B, EG Tech., Anyang, Korea).

In order to examine components of the road dust, all of the particles in 10 mg road dust of 212–500 μm (sample code: RD1) were classified into the kinds with the naked eye under a microscope. Rough density of each particle was also measured using the salt solutions of 1.2 and 1.6 g/cm^3^. The road dust samples of 212–500 μm (20 mg, sample code: RD2), 106–212 μm (10 mg, sample code: RD3), and 63–106 μm (5 mg, sample code: RD4) were used.

### 2.4. Density Separation Method

Approximately 50 mL of salt solution or chloroform was put into a glass container (diameter 70 mm, height 40 mm), and the road dust sample was put into the container. In the case of using chloroform, the sonication (ultrasonic cleaner, Jeiotech Co., Daejeon, Korea) process was performed for an additional 10 min. In order to sufficiently wet the particles, the solution was dropped several times with a micro pipette and left for 30 min. After this process was repeated twice, the particles sunk and floated were separated using a micro pipette. The separated particles were transferred to a petri dish, washed 3 times with deionized water and finally washed once more with ethanol, and then dried at room temperature.

### 2.5. Characterization of TRWPs

Morphology analysis of the classified TRWPs was performed using a digital microscope (Leica DM4 M, Leica Microsystems, Wetzlar, Germany). Lengths of the major and minor axes were measured using the 2D analysis module, and the aspect ratio was obtained. Other tiny particles deposited on the TRWP surface were also observed.

Rubber composition of the TRWPs was analyzed using pyrolysis-gas chromatography (Py-GC). YL6500GC (Younglin Co., Seoul, Korea) equipped with CDS 2000 Pyroprobe (CDS Analytical Inc., Oxford, PA, USA) was used. The interface temperature of the pyrolyzer was 250 °C. The sample was put in a quartz tube and both sides were covered with glass wool and then pyrolyzed at 520 °C for 10 s. Temperature of the GC injector was 250 °C. An HP-5 column (30 m × 0.25 mm, 0.25 μm thickness) was used. The split ratio was 1:15 and N_2_ was used as the carrier gas. The GC oven temperature program was as follows: 30 °C (held for 3 min) to 180 °C (held for 1 min) at 10 °C/min, then to 250 °C (held for 3 min) at 10 °C/min.

## 3. Results and Discussion

### 3.1. Composition and Density Properties of the Road Dust

In order to examine the kinds of components in the road dust, each particle in 10 mg road dust of 212–500 μm was classified and their densities were measured. Among the black particles, elastic particles with elongated or round shapes were classified into TRWPs. On the other hand, APWPs were selected based on the feature of being inelastic and breaking when pressed.

Besides TRWPs and APWPs, there were various kinds of particles in the road dust samples such as MPs, GPs, GBs, RPWPs, PRPs, and PPs as listed in Table 1. MPs were most abundant in the road dust, and they had a rigid and angular shape. PPs were thin and slightly hard. RPWPs were particles produced by wearing road marking paint, and they were white or yellow color and crumbled when pressed. PRPs were generated from plants around the bus stop, and they could be crushed into smaller pieces. GPs were transparent and angular shape, whereas GBs were spherical.

Rough densities of the classified particles of the RD1 sample are summarized in Figure 1. Densities of TRWPs, RPWPs, and PRPs were widely spread over 1.2 g/cm^3^, while those of APWPs, MPs, GPs, and GBs were over 1.6 g/cm^3^. Density of PPs was lower than 1.6 g/cm^3^. Based on these results, the reference density was set to 1.6 g/cm^3^. When the road dust sample is put in the matrix solution with density of 1.6 g/cm^3^, PPs will be floated while MPs, APWPs, GPs, and GBs sink. For TRWPs, RPWPs, and RPPs, some of them can float and the others may sink.

In order to examine the density distribution of the TRWPs by 0.10 g/cm^3^ unit, 10 TRWPs were randomly selected from the road dust of 106–212 μm because there were rare TRWPs in the road dust of 212–500 μm. The TRWPs had densities of 1.20–1.60 g/cm^3^, and those with low density of 1.20–1.30 g/cm^3^ were much greater than the others, as shown in Figure 2. Based on the density ranges of TRWPs, the density separation was carried out using the salt solutions with densities of 1.20, 1.30, 1.40, 1.50, 1.60, and 1.70 g/cm^3^.

### 3.2. Differentiation of TRWPs and APWPs

Shapes and colors of TRWP and APWP in road dust are very similar, and some TRWPs with high density are the same density region of APWPs. For this reason, it is hard to differentiate TRWPs from APWPs in the same density range. Fortunately, bitumen used as a binder of asphalt pavement is easily dissolved with some solvents such as chloroform, but crosslinked rubber does not dissolved with any solvents. In this study, TRWPs were differentiated from APWPs in the same density range using chloroform treatment. When an APWP is put into chloroform, bitumen part of the APWP is dissolved, as shown in Figure 3.

A TRWP and an APWP with similar size and shape were selected, and chloroform was dropped on them (Figure 4). After the chloroform treatment, the TRWP still maintained its initial shape but the APWP split into pieces and the degree was more severe by increasing the chloroform drop. When the whole bitumen part of an APWP was totally dissolved in chloroform, the APWP shape disappeared and aggregate (broken stone) in the APWP was appeared. Hence, we can conclude that APWPs mixed with TRWPs are removed by chloroform treatment. By the chloroform treatment, binder components such as bitumen in APWPs dissolve and the stone components with high density settle down. 

### 3.3. Density Separation of the Road Dust Samples

An experimental procedure for TRWP separation from road dust and classification of the TRWPs by density is described in Figure 5. First, the road dust sample was put into the salt solution of 1.60 g/cm^3^ and divided into high- and low-density particles corresponding to the sunk and floating parts, respectively. Second, the high-density moiety was put into the salt solution of 1.70 g/cm^3^, the floated 1 was treated with chloroform to remove APWPs, and TRWPs were selected from the mixture. The TRWPs selected from the second step have density of 1.60–1.70 g/cm^3^. Third, the moiety floated in the salt solution of 1.60 g/cm^3^ was put into the salt solution of 1.20 g/cm^3^ to remove light particles such as some of PRPs and PPs. Fourth, the moiety sunk in the salt solution of 1.20 g/cm^3^ was put into the salt solution of 1.40 g/cm^3^ and the sunk and floated parts were put into the salt solutions of 1.50 and 1.30 g/cm^3^, respectively. Fifth, the moieties sunk and floated in the salt solution of 1.50 g/cm^3^ were divided into 2 groups of 1.50–1.60 and 1.40–1.50 g/cm^3^, respectively. The TRWPs selected from the fifth step have densities of 1.50–1.60 and 1.40–1.50 g/cm^3^, respectively. Finally, the moieties sunk and floated in the salt solution of 1.30 g/cm^3^ were divided into 2 groups of 1.30–1.40 and 1.20–1.30 g/cm^3^, respectively. The TRWPs selected from the final step have densities of 1.30–1.40 and 1.20–1.30 g/cm^3^, respectively.

The numbers of TRWPs after the density separation were counted and the results have been summarized in Figure 6. There was no TRWP below 1.20 g/cm^3^ or above 1.70 g/cm^3^. For the RD2 sample of 212–500 μm, TRWPs were rarely found and another road dust sample of 212–500 μm did not have any TRWP. There were only TRWPs with relatively low densities below 1.50 g/cm^3^ in the RD2 sample. For the RD3 sample of 106–212 μm, TRWPs with all density ranges of 1.20–1.70 g/cm^3^ were found. There were relatively even TRWPs with density ranges of 1.30–1.70 g/cm^3^ in the RD4 sample of 63–106 μm. By decreasing the road dust size, the TRWP size tended to smaller.

The contents of the TRWPs in the road dust samples are summarized in Figure 7. The TRWP content clearly increased as the road dust sample size decreased. This means that size distribution of TWPs produced by abrasion of tire treads shifted to smaller one or initial TRWPs were broken into smaller pieces by friction between the road and the tire. For the RD2 sample of 212–500 μm, content of the TRWP with 1.20–1.30 g/cm^3^ was much greater than that with 1.40–1.50 g/cm^3^. For the RD3 sample of 106–212 μm, content of the TRWP with 1.40–1.50 g/cm^3^ was larger than those of the other sizes. Compared to the TRWPs with the same density range, the TRWP content tended to increase with decrease in the particle size.

### 3.4. Characterization of TRWPs Classified by the Density Separation

Shapes of the classified TRWPs were analyzed and the aspect ratios were measured. Magnified images of the representative TRWPs are shown in Figure 8. The TRWP surface was magnified to examine the tiny particles attached to the surface as shown in Figure 9. Particle sizes on the TRWP surface varied and the number of particles on the TRWP surface tended to increase as the TRWP density increased.

There were rare TRWPs in the RD2 sample (212–500 μm), but their aspect ratios were very different from each other, as listed in Table 2. Only 3 TRWPs were found in the RD2 sample. The L3 TRWP had very long shape and its aspect ratio was over 30. A total of 21 TRWPs were found in the RD3 sample (106–212 μm) and the TRWPs were given a number according to the aspect ratio as well as the density (Table 3). There was no specific trend for the aspect ratios with the density. The lowest aspect ratio was 1.22, while the highest one was 5.21. There were many TRWPs with very low aspect ratios below 2.00 g/cm^3^ in the TRWPs of 1.60–1.70 g/cm^3^. 65 TRWPs were found in the RD4 sample (63–106 μm) and they were given a number according to the aspect ratio as well as the density (Table 4). The lowest aspect ratio was 1.21, while the highest one was 4.31. Aspect ratios of the TRWPs in the RD4 sample were relatively lower than those in the RD3 sample. There were many TRWPs with very low aspect ratios below 2.00 for all of the density ranges. Ratios of the TRWPs with very low aspect ratios below 2.00 tended to increase by increasing the density. The ratios were 31, 42, 35, and 50% for the TRWPs with the densities of 1.30–1.40, 1.40–1.50, 1.50–1.60, and 1.60–1.70 g/cm^3^, respectively.

Py-GC chromatograms of the TRWPs separated from the RD2, RD3, and RD4 samples showed similar pyrolysis products, as shown in Figure 10. The major pyrolysis products were isoprene and dipentene which are the key pyrolysis products of natural rubber (NR) [13,14,15,16]. Since bus tire treads are mainly made of NR [17,18,19,20], the TRWPs should come from abrasion of bus tire treads. Styrene was one of main pyrolysis products of the TRWPs and this could originate from abrasion of tire tread made of styrene-butadiene rubber (SBR) or APWPs attatched to the TRWPs. Styrene and 4-vinylcyclohexene (VCH) are the key pyrolysis products of SBR [21,22,23]. VCH is also the key pyrolysis products of butadiene rubber (BR) [24]. Hence, the TRWPs should be composed of NR/SBR biblend or NR/BR/SBR triblend compound. The TRWP of the RD2 sample showed very intensive styrene peak but the VCH peak was relatively small. This implies that some styrene might come from the other source such as APWP not SBR. For the TRWP of the RD3 sample, the peak intensities of styrene and VCH were nearly the same, which means that BR content in the TRWPs was not lower than SBR sample. 13 TRWPs of the RD3 sample were analyzed together so their rubber compositions might be different each other.

## 4. Conclusions

Based on the density ranges of TRWPs, the density separation was carried out using the salt solutions with different densities of 1.20, 1.30, 1.40, 1.50, 1.60, and 1.70 g/cm^3^. TRWPs were differentiated from APWPs in the same density range using chloroform treatment. TRWPs were classified into 5 groups according to the density ranges of 1.20–1.30, 1.30–1.40, 1.40–1.50, 1.50–1.60, and 1.60–1.70 g/cm^3^. There was no TRWP below 1.20 g/cm^3^ or above 1.70 g/cm^3^. For the RD2 sample (212–500 μm), TRWPs were rarely found. For the RD3 sample (106–212 μm), TRWPs with all density ranges of 1.20–1.70 g/cm^3^ were found. There were relatively even TRWPs with density ranges of 1.30–1.70 g/cm^3^ in the RD4 sample (63–106 μm). For the TRWPs with the same density range, the TRWP content tended to increase with decrease in the particle size. Sizes of the tiny particles adsorbed on the TRWP surface varied and the number of the tiny particles tended to increase as the TRWP density increased. There was no specific trend for the aspect ratios with the TRWP density. Almost TRWPs were in the aspect ratio range of 1.2–5.2. Percentage of the TRWPs with very low aspect ratios below 2.00 tended to increase by increasing the TRWP density. Major pyrolysis products of the TRWPs were isoprene and dipentene which means that the TRWPs were mainly made of NR and they came from the abrasion of bus tire treads. Styrene and VCH were also observed as the main pyrolysis products, meaning that the TRWP might be composed of NR/SBR biblend or NR/BR/SBR triblend compound.

## Figures and Tables

**Figure 1 polymers-14-01005-f001:**
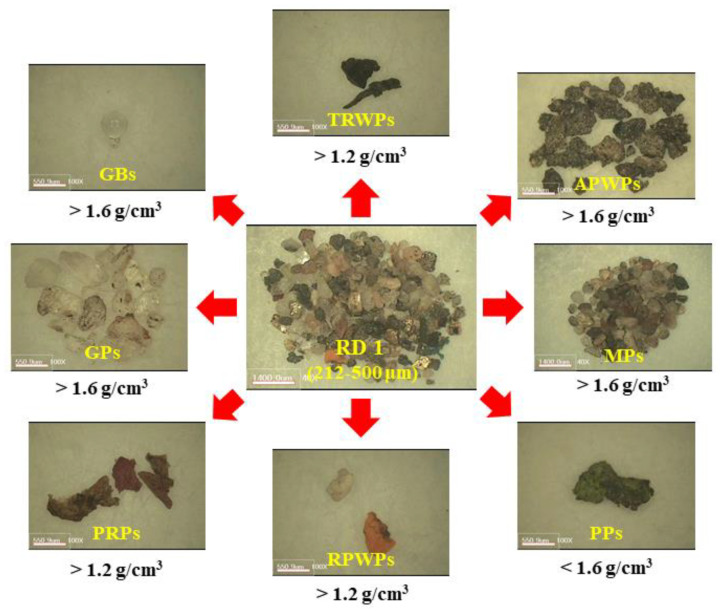
Density ranges of various particles in the RD1 sample. The scale bar of the center image is 1400 μm, and those of the others are 551 µm.

**Figure 2 polymers-14-01005-f002:**
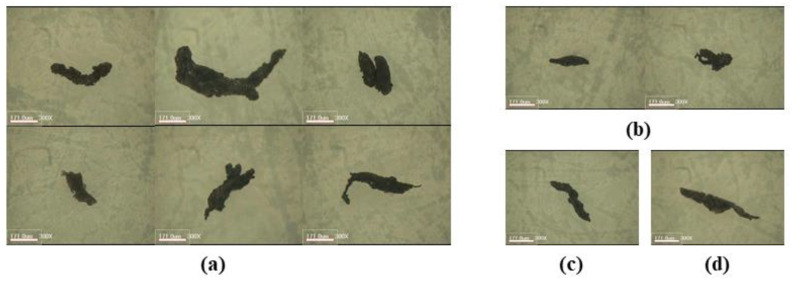
Magnified images of TRWPs selected from the road dust of 106–212 μm. (**a**) 1.20–1.30 g/cm^3^, (**b**) 1.30–1.40 g/cm^3^, (**c**) 1.40–1.50 g/cm^3^, and (**d**) 1.50–1.60 g/cm^3^. The scale bars are 171 μm.

**Figure 3 polymers-14-01005-f003:**
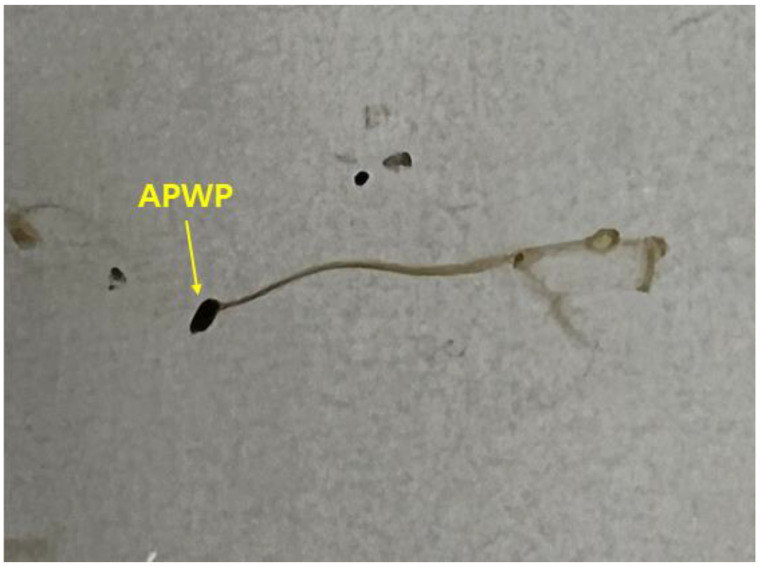
Dissolution of bitumen in APWP by chloroform.

**Figure 4 polymers-14-01005-f004:**
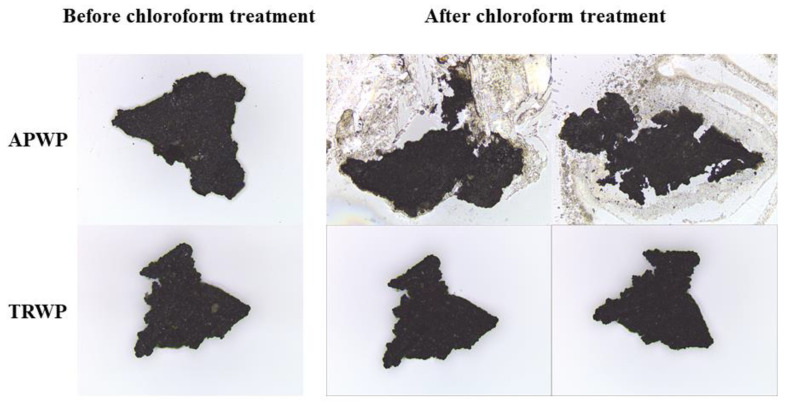
Difference in the shape changes of TRWP and APWP selected from the road dust of 212–500 µm after the chloroform treatment.

**Figure 5 polymers-14-01005-f005:**
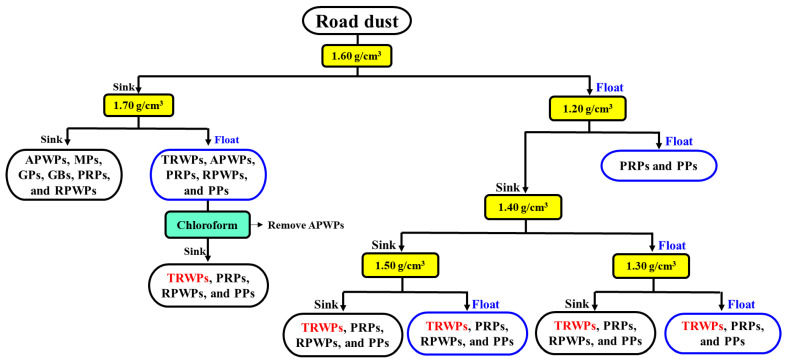
Separation procedure of TRWPs from road dust by the density difference.

**Figure 6 polymers-14-01005-f006:**
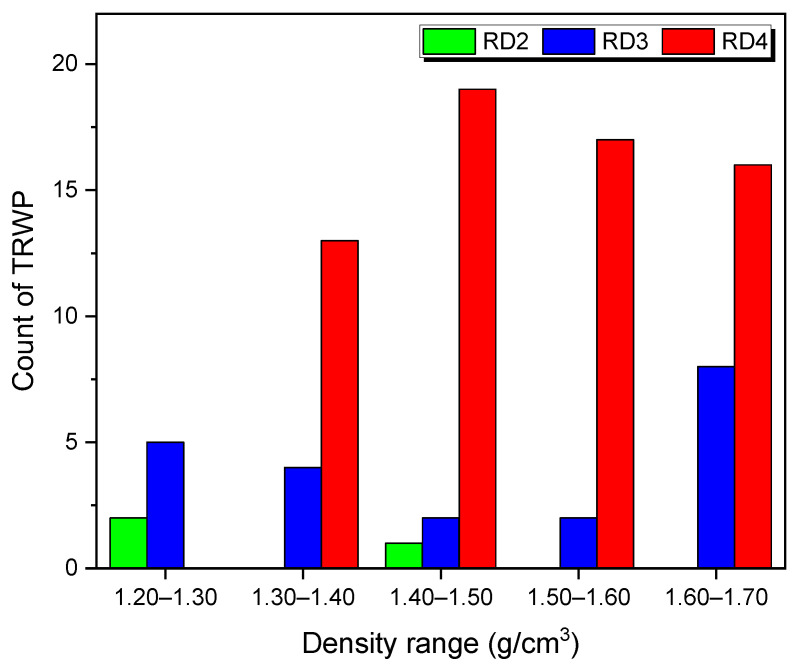
Variations in the numbers of TRWPs in the road dust samples with the TRWP density.

**Figure 7 polymers-14-01005-f007:**
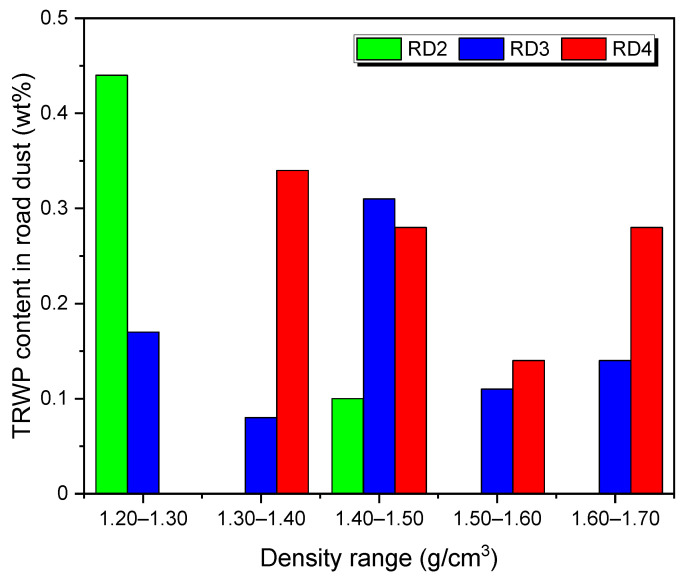
Variations in the TRWP contents in the road dust samples with the TRWP density.

**Figure 8 polymers-14-01005-f008:**
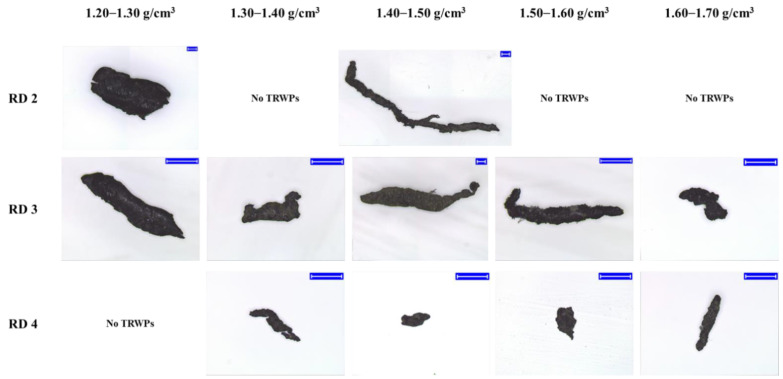
Magnified images of the TRWPs classified from the road samples. The blue scale bar is 100 µm.

**Figure 9 polymers-14-01005-f009:**
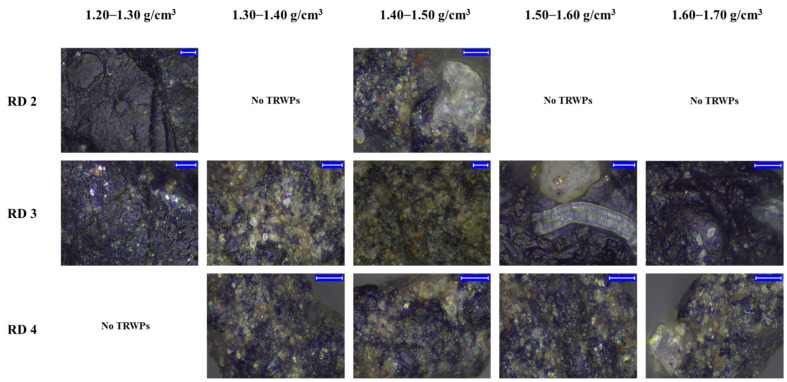
Magnified surface images of the TRWPs classified from the road samples. The blue scale bar is 10 µm.

**Figure 10 polymers-14-01005-f010:**
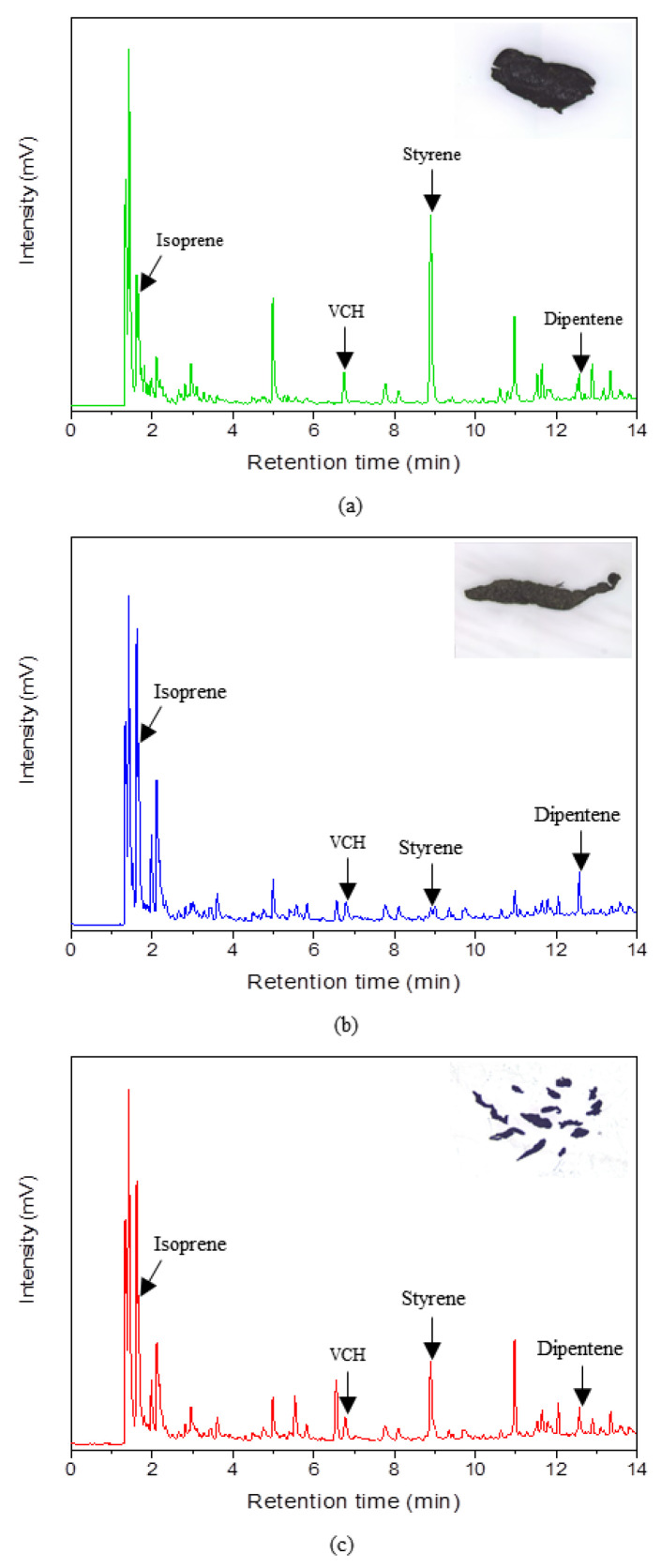
Py-GC chromatograms of the TRWPs classified from the road dust samples. (**a**) the RD2 sample (single TRWP, 1.20–1.30 g/cm^3^), (**b**) the RD3 sample (single TRWP, 1.40–1.50 g/cm^3^), and (**c**) the RD4 sample (13 TRWPs, 1.30–1.40 g/cm^3^).

**Table 1 polymers-14-01005-t001:** Particle components of the RD1 sample (212–500 μm).

Particle	Ratio (wt%)
Tire-road wear particles (TRWPs)	0.5
Asphalt pavement wear particles (APWPs)	7.0
Mineral particles (MPs)	77.2
Glass particles (GPs)	11.7
Glass beads (GBs)	1.5
Road paint wear particles (RPWPs)	1.1
Plant-related particles (PRPs)	0.9
Plastic particles (PPs)	0.1

**Table 2 polymers-14-01005-t002:** Aspect ratios of TRWPs separated from the RD2 sample (212–500 μm).

Density (g/cm^3^)	TRWP No.	Length (µm)	Width (µm)	Aspect Ratio
1.20–1.30	L1	906.818	443.243	2.046
	L2	819.610	328.868	2.492
1.40–1.50	L3	2210.560	70.730	31.253

**Table 3 polymers-14-01005-t003:** Aspect ratios of TRWPs separated from the RD3 sample (106–212 μm).

Density (g/cm^3^)	TRWP No.	Length (µm)	Width (µm)	Aspect Ratio
1.20–1.30	M1	213.525	102.610	2.081
	M2	216.658	79.177	2.736
	M3	288.630	101.299	2.849
	M4	212.629	64.337	3.305
	M5	388.050	100.524	3.860
1.30–1.40	M6	81.069	49.217	1.647
	M7	97.367	45.208	2.154
	M8	207.638	91.421	2.271
	M9	354.829	68.065	5.213
1.40–1.50	M10	142.078	57.064	2.490
	M11	744.581	189.547	3.928
1.50–1.60	M12	176.207	97.506	1.807
	M13	387.485	104.674	3.702
1.60–1.70	M14	84.341	69.123	1.220
	M15	119.108	68.880	1.729
	M16	135.348	76.499	1.769
	M17	155.191	86.731	1.789
	M18	173.171	86.788	1.995
	M19	167.619	69.844	2.400
	M20	152.797	48.497	3.151
	M21	166.919	43.192	3.865

**Table 4 polymers-14-01005-t004:** Aspect ratios of TRWPs separated from the RD4 sample (63–106 μm).

Density (g/cm^3^)	TRWP No.	Length (µm)	Width (µm)	Aspect Ratio
1.30–1.40	S1	75.987	62.774	1.210
	S2	84.482	69.276	1.219
	S3	104.769	66.094	1.585
	S4	102.466	58.350	1.756
	S5	85.681	40.413	2.120
	S6	89.423	41.858	2.136
	S7	143.217	64.288	2.228
	S8	106.022	42.768	2.479
	S9	175.500	67.418	2.603
	S10	132.855	47.826	2.778
	S11	122.079	42.096	2.900
	S12	181.297	51.182	3.542
	S13	185.188	44.504	4.161
1.40–1.50	S14	68.380	51.267	1.334
	S15	51.300	36.674	1.399
	S16	74.494	50.048	1.488
	S17	70.844	46.010	1.540
	S18	79.489	50.976	1.559
	S19	71.347	45.476	1.569
	S20	70.813	44.320	1.598
	S21	74.040	37.798	1.959
	S22	96.020	46.071	2.084
	S23	87.544	41.531	2.108
	S24	75.285	34.674	2.171
	S25	95.998	43.113	2.227
	S26	88.719	38.018	2.334
	S27	69.131	26.746	2.585
	S28	92.232	30.691	3.005
	S29	87.772	27.016	3.249
	S30	160.384	41.452	3.869
	S31	244.556	62.589	3.907
	S32	75.939	17.631	4.307
1.50–1.60	S33	78.593	53.737	1.463
	S34	104.537	70.370	1.486
	S35	58.257	37.340	1.560
	S36	110.743	65.747	1.684
	S37	71.824	39.602	1.814
	S38	80.917	41.093	1.969
	S39	76.901	31.860	2.414
	S40	93.174	36.110	2.580
	S41	159.109	54.829	2.902
	S42	126.799	42.858	2.959
	S43	150.389	48.767	3.084
	S44	112.468	34.436	3.266
	S45	139.460	40.920	3.408
	S46	111.306	32.553	3.419
	S47	105.397	30.033	3.509
	S48	121.955	34.548	3.530
	S49	168.570	47.300	3.564
1.60–1.70	S50	150.119	106.791	1.406
	S51	100.285	68.619	1.461
	S52	109.567	72.483	1.512
	S53	210.149	131.425	1.599
	S54	183.867	107.128	1.716
	S55	99.901	53.580	1.865
	S56	82.631	42.693	1.935
	S57	97.152	48.720	1.994
	S58	152.058	74.227	2.049
	S59	107.015	50.644	2.113
	S60	194.690	84.565	2.302
	S61	83.901	35.866	2.339
	S62	127.199	49.838	2.552
	S63	123.673	48.075	2.573
	S64	285.707	75.658	3.776
	S65	364.601	91.268	3.995

## Data Availability

The data presented in this study are available on request from the corresponding author.

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
