# Peer review of "Classification and Characterization of Tire-Road Wear Particles in Road Dust by Density"

_polymers, 2022, doi:10.3390/polym14051005_

Round 1

Reviewer 1 Report

Dear authors,

comments are in the attachment.

Author Response

Q1. Lines 68, 75, and 77: Maybe some other properties of used materials?

A1. Purities of the sodium bromide and sodium iodide were added in Line 68. Densities of the saturated solutions of sodium bromide and sodium iodide were given in Lines 74 and 77, respectively.

Q2. Line 85: Is it really possible to see a particle in micrometer with the naked eye?

A2. The kinds of particles were differentiated with the naked eye under a microscope. “under a microscope” was added Line 86.

Q3. Line 100: Maybe some other method of characterization will be useful for example FTIR?

A3. We characterized a single TRWP in terms of the morphology and the rubber composition. The morphology was analyzed using a digital microscope and the rubber composition was analyzed using Py-GC. TWP is carbon black-filled rubber compound, and TRWP is composed of TWP and various other particles such as stone powder and asphalt pavement wear particles. The carbon black in TWP can interfere with the rubber peaks of FTIR analysis. The components covered with the TRWP can also interfere with FTIR analysis of rubber composition of TWP.

Q4. Table 1: It is not clear how authors were calculated the ratio of some components?

A4. Each particle in the RD1 sample was differentiated with the naked eye under a microscope as shown in Figure 1, and the each component was weighed.

Q5. Line 129: how is determined the density of individual components?

A5. The density of individual component was determined by the density separation method using the salt solutions with different densities of 1.2 and 1.6 g/cm3. “using the salt solutions of 1.2 and 1.6 g/cm3” was added Lines 86-87.

Reviewer 2 Report

The English is not of a high enough standard to publish as is and requires a thorough edit from a more proficient English writer. This is beyond the role of a reviewer

Methodologically there are limitations - for instance the method of collection by sweeping will bias the material towards the very coarse fraction and makes the results less applicable to the air quality community and this should be  acknowledged in any future version. 

Some of the methods (e.g. the chloroform treatment) are not well written and appear observational rather than quantitative and repeatable. The description of how the different sources (MPs, GPs, GBs, RPWPs, PRPs, and PPs) appeared very subjective.

Author Response

Q1. The English is not of a high enough standard to publish as is and requires a thorough edit from a more proficient English writer.

A1. The corrections were marked in blue.

Q2. Methodologically there are limitations - for instance the method of collection by sweeping will bias the material towards the very coarse fraction and makes the results less applicable to the air quality community and this should be acknowledged in any future version. Some of the methods (e.g. the chloroform treatment) are not well written and appear observational rather than quantitative and repeatable. The description of how the different sources (MPs, GPs, GBs, RPWPs, PRPs, and PPs) appeared very subjective.

A2. This method can be useful for separation of TRWPs from road dust. TRWPs generated on the road generally less than 500 μm. We agree with your comment that the method of collection by sweeping will bias the material towards the coarse fraction. Purpose of this method is separation of TRWPs in the environmental samples from the other particles. Sampling methods will be various and they can have pros and cons. We believe that TRWPs in a certain environmental sample can be easily separated by this method irrespective of the sampling methods.

The chloroform treatment was used for removing APWPs from the mixture sample with the same density range. Shapes and colors of TRWP and APWP in road dust are very similar, and some TRWPs with high density are the same density region of APWPs. For this reason, it is hard to differentiate TRWPs from APWPs in the same density range. By the chloroform treatment, binder components in an APWP such as bitumen are dissolved and the stone components with high density are settled down. Since TWPs are crosslinked rubbers, they are not dissolved in chloroform. Dissolution of bitumen in an APWP by chloroform was shown in Figure 3. Shape changes of TRWP and APWP after the chloroform treatment were shown in Figure 4. The sentence “By the chloroform treatment, binder components such as bitumen in APWPs are dissolved and the stone components with high density are settled down.” was added in Lines 161-163.

             As shown in Figure 1, shapes of the MPs, GPs, GBs, RPWPs, PRPs, and PPs were different each other. Classification of the particles was objective because their properties are different. MPs are stone powder, GPs are grinded glass, GBs are glass beads for improvement of retro-reflection, RPWPs are wear particles of painted road lines, PRPs are small particles of bark, wood, and leaf, and PPs are plastic particles. Their visual characteristics were described in Lines 121-126.

Round 2

Reviewer 1 Report

Dear authors,

all comments are accepted and manuscript is now acceptable for publication.